# Did Covid-19 make things worse? The pandemic as a push factor stimulating the emigration intentions of junior doctors from Poland: A mixed methods study

Dominika Pszczółkowska[1,2][☯]*, Sara Bojarczuk[1][☯], Maciej Duszczyk[1,2][☯], Kamil Matuszczyk[1,2][☯], Emilia Szyszkowska[3][☯]

1 Centre of Migration Research, University of Warsaw, Warszawa, Poland, 2 Faculty of Political Science and International Studies, University of Warsaw, Warszawa, Poland, 3 Centre of Migration Research, University of Warsaw and Warsaw School of Economics, Warszawa, Poland

☯ These authors contributed equally to this work.
* d.pszczolkows2@uw.edu.pl

**Data Availability Statement:** Quantitative data is available in a repository at https://osf.io/vkbnx/ Qualitative data will not be made available because

## Abstract

Covid-19 has challenged health systems around the world and increased the global competition for medical professionals. This article investigates if the pandemic and its management became an important push factor influencing the migration intentions of medical students and junior doctors and how this factor compared in importance to others. A mixed methods study–a survey and in-depth interviews–was conducted with final-year students at public medical universities in Poland, a country already suffering from a significant emigration of medical staff. The research demonstrated that the difficulties of the Polish healthcare system in dealing with Covid-19 were a factor that slightly positively influenced the emigration intentions of medical students and junior doctors. Nevertheless, the pandemic's influence was not decisive. Factors such as the socio-political situation in Poland (.440**) (including hate speech directed at doctors by politicians and patients), the participants' family situation (.397**), healthcare system organization (.376**), or the opportunity of pursuing a planned career path (.368**) proved more influential. Salary is still important but did not turn out to be among the decisive factors. This allows us to conclude that migration decisions of medical students have a very well-established basis that does not fundamentally change even under the influence of such dramatic situations as the pandemic. This conclusion has important implications for healthcare management and the ongoing discussion in migration studies on the evolution of push and pull factors in place and time.

## Introduction

Healthcare workers are one of the most geographically mobile professional groups, who also migrate between developed countries [1–4]. Europe's ageing population creates new opportunities for doctors' employment in countries with better prospects and standards of living,

due to the large amount of personal data of the respondents they cannot be fully anonymised.

**Funding:** This paper was written as part of the research project, "Migration plans of medical students and their implementation. Will they really leave?", financed by the National Science Centre of Poland, within the OPUS programme (contract no: UMO-2020/39/B/HS5/00464). The funders had no role in study design, data collection and analysis, decision to publish, or preparation of the manuscript.

**Competing interests:** The authors have declared that no competing interests exist.

facilitated by the extensive European Union professional qualification recognition system. The World Health Organization estimates that there will be a global shortage of around 15 million workers in the healthcare sector by 2030 [5], which will increase the competition for medical professionals. While much is known about the factors and circumstances prompting health-care workers to migrate, there is a paucity of knowledge about the impact of emergencies, including pandemic-driven health crises, on intentions to leave or stay in a particular country.

The Covid-19 pandemic has challenged health systems around the world, revealing shortages of employees and proving how dependent OECD countries are on migrant workers [6]. To attract skilled professionals, who were not eager to migrate during the initial months of the pandemic [7], many high-income countries facilitated access to healthcare employment for foreign-trained personnel, for example by reducing the timeframes for recognition of professional qualifications or lowering the required level of language proficiency [5, 8, 9]. Healthcare workers were found to have been attracted by countries providing better working conditions and which had proven resilient during the pandemic [10].

As Poland has been facing a systemic shortage of medical staff for many years, the outflow of doctors poses a major challenge for safeguarding the proper functioning of the healthcare system. The number of doctors per 1,000 citizens (3.4), as well as the number of medical graduates in Poland, are below the average for both the EU and OECD countries [11]. In most countries, the number of practising doctors has increased over the last two decades, while in Poland this increase was minor [12]. Also, since EU accession in 2004, the number of Polish and other Central European medical graduates seeking to pursue careers abroad has increased [13–15]. The results of two surveys conducted among medical students in 2008 and 2011 indicated a continuous interest in emigration due to a lack of opportunities for professional development in Poland, and the intention to search for knowledge, better salaries, and the prestige of the profession outside of Poland [16–18]. Recently, the migration rate of doctors only to other EU countries was estimated at 7% [19], while the inflow of foreign-born doctors until the outbreak of war in Ukraine was marginal [20].

The Covid-19 pandemic has exposed the weaknesses of the Polish healthcare system, with the country recording one of the highest rates of excess deaths per million among OECD countries [12]. Studies demonstrate the system's shortcomings in response to the challenges of the pandemic [21, 22] and medical staff's dissatisfaction with the level of funding and salaries [23]. In such circumstances, it is crucial to examine the extent to which the pandemic was significant for pushing doctors out of Poland. The objectives of this study are thus to estimate the prevalence of migration intentions of final year medical students, as well as to examine the factors influencing such intentions—both sudden, represented by the Covid-19 pandemic, and steady, such as economic and socio-political factors.

The research is based on a mixed methods design consisting of two components: an online survey among medical students (several months before their graduation) and in-depth interviews conducted with them a few months later, when they were already junior doctors. We make an original contribution to the growing body of research on the factors and circumstances that shape the mobility trajectories of health professionals [4]. Our findings advance the discussion on the importance of the impact of extraordinary events or circumstances (such as a pandemic) on the migration decisions of high-skilled personnel. From an empirical point of view, we enrich the discussion on the career paths of junior doctors in sending countries, especially in the Central and Eastern European region [24, 25].

The article is structured as follows: we begin by presenting our theoretical framework and reviewing the literature on how the Covid-19 pandemic acted as a push-pull factor in the migration of health professionals around the world. We then focus on factors that push junior doctors out of Poland. Based on the available empirical evidence, we formulate two

hypotheses. Following the outline of a mixed methods approach, we discuss the results of two logistic regression models, enriched by qualitative analysis, which lead to our conclusions.

In this article, we use the notion of 'migration intentions' as a concept referring to migrations not necessarily already carried out but considered as a serious option. The term is widely used in the literature on the migration of doctors [26–28]. We refer to 'medical students' who have not yet graduated, and to 'junior doctors', meaning all those who have graduated but have not completed their specialisation and reached consultant or general practitioner level [29].

## Theoretical background: The push–pull approach

Theoretically, the study is based on the classical approach to explaining decision-making processes through an analysis of push and pull factors, as proposed by Everett S. Lee [30], and extensively developed by later authors [31–34]. It assumes that decisions to migrate are free and boundedly rational, based on criteria which push a given person out from the state of origin and pull them to the receiving state. Also important are intervening factors, such as geographical distance and legal regulations (in the case of doctors, this concerns especially the recognition of professional qualifications), and personal factors. The value ascribed by an individual to a material or non-material good (such as a house, proximity of an airport, or appropriate schools for the children), may depend on their demographic features, education, or individual perception. Both positive and negative factors must be considered in the origin and the potential destination [35, 36]. Akl et al. [37], in their study of Lebanese medical professionals, propose the notions of 'retain' factors at the origin, and 'repel' factors at the destination, in addition to pull and push factors. Carling and Bivand Erdal [38] have also reverted to the notions of 'push/retain' on one side and 'pull/repel' on the other.

A review of the recent literature (up to 2023) on factors influencing doctors' migration intentions indicated the desire for a better quality of life, career and training opportunities, as well as financial gain as the strongest factors influencing such decisions [27]. For most graduates, factors such as the level of salaries, work conditions, possibilities of professional development, employment security and stability, as well as linguistic and other cultural factors are frequent push or pull factors [39]. Additional factors may include equipment at the workplace, organization of the healthcare system, working hours, work atmosphere, stress levels or treatment, the esteem for doctors in a given country or employment conditions [14, 40]. In the case of highly skilled workers, such as doctors and nurses, the determinants of their decisions and aspirations to migrate are significantly different from other categories of mobile workers. Research among healthcare workers in Romania reveals a surprising relationship, according to which the higher the satisfaction with earnings in the country of origin, the greater the temptation to work in another country [41].

'Retain' factors in origin may include the possibility of gaining additional wages in private healthcare, possibilities of professional advancement and prestige (which are sometimes easier to achieve in one's country of origin), or factors related to personal lives, such as the presence of family. 'Repel' factors may be linked with the 'glass ceiling', or difficulty of advancing professionally due to racial discrimination or linguistic difficulties. Highly skilled migrants, including medical staff, also take into account the degree of political stability in a country or opportunities for self-development [2]. In addition, findings from Polish research suggest that younger doctors (i.e., residents) are more likely to leave than specialist doctors [26]. In light of the Covid-19 pandemic, it is important to consider the issue of how sudden factors, such as a health emergency, war, or natural catastrophe influence migration intentions, and the relative importance of factors taken into consideration.

## The pandemic as a push factor in healthcare workers' migration

While numerous studies provide evidence of the impact of the Covid-19 pandemic on mobility and migration [42], in particular the deterioration of the situation of migrant workers and the emergence of border crossing barriers [43, 44], little is still known about how this global health crisis influenced future migration intentions. According to the Eurobarometer survey [45], the Covid-19 pandemic had a minimal impact on Europeans' mobility plans. Only 2% of respondents postponed their plans of working abroad, 3% were less convinced of the idea of going abroad, and a further 2% abandoned their plans. Recent research in Hong Kong shows that the severity of a pandemic in a country (manifested by high levels of mortality and morbidity) had a positive effect on increasing the intentions to migrate abroad, especially among young, well-educated people [46].

Healthcare workers are a special case because the Covid-19 pandemic has underscored their key role in public health services and affected their mobility in particular ways. Although much has been published on the experience of healthcare professionals working in hospitals during the pandemic [47–50], the issue of the impact of the pandemic on doctors' migration decisions is inconclusive. To date, the literature suggests—on the one hand—that medical staff are more likely to postpone migration in a pandemic, in part due to the health risks linked with travelling and undertaking work in a new environment in such circumstances [51], but on the other hand that state policies aiming to attract medical staff have a significant impact on migration levels. Many such policies were implemented in reaction to the pandemic [52]. In studies suggesting an increase of the migration potential of healthcare workers, the pandemic was not the main motive of this increase, but rather a factor exposing weak healthcare systems [53, 54] and strengthening previous migration intentions caused by other, stronger, push factors [55–57]. It is noteworthy that, irrespective of the doctors' country of origin, the main drivers of emigration during the pandemic were the belief in better career opportunities, higher quality equipment and access to medical facilities, followed by factors related to employment conditions like salary, working hours, type of contract [19, 55, 58]. For medical students, the deterioration of the quality of education (i.e., online courses) and the desire for professional development abroad were important.

In Poland, studies and official data suggest that the impact of the pandemic on the migration potential of doctors may have been strong, albeit delayed. Data show an initial decline in the number of issued certificates of qualifications (which are needed to work abroad). This was likely due to mobility restrictions [59]. By contrast, since the beginning of 2022, the Supreme Chamber of Doctors has been warning of an unprecedentedly high number of certificate applications, which suggests a growing tendency to emigrate [60]. The latest figures show that almost 1,000 doctors were issued the certificates in 2022, which was more than in previous years [61]. In a study of nearly 3,000 active healthcare workers, only one in three respondents stated that the pandemic had no direct impact on their plans related to their profession (34%), while one in ten respondents was thinking of leaving their job (12%) or moving abroad (12%) [62].

## Hypotheses and methods

**Hypotheses.**   Despite Poland transforming rapidly from a country of emigration into a country of immigration (which is also influenced by the arrival of war refugees from Ukraine) [63], in the case of medical personnel, as we indicated above, there are still far more people emigrating from Poland than arriving.

At the same time, doctors' working conditions and quality of life deteriorated during the pandemic. Studies around the world have shown that caring for patients with Covid-19 not

only led to an increase in infections among frontline healthcare workers but also affected their mental health, causing anxiety, depression and job burnout [47–50, 64–66]. In Poland, deaths of more than 400 healthcare workers due to the pandemic were reported by mid-2021 [67]. In a survey of medical students, more than 80% of respondents indicated that the public's dislike and distrust of doctors increased significantly during the pandemic [68]. Some said this lowered their enthusiasm for a medical career in Poland [68]. Also, a growing problem of hate speech against doctors has been noted, as well as physical attacks or problems involving family members of medical personnel (e.g., refusal to admit a doctor's child to kindergarten) [62, 69]. Therefore, in our study, we hypothesize that:

H1: *The Covid-19 pandemic has increased the migration intentions of medical students and junior doctors in Poland.*

*Given that r*esearch results from other countries suggested that the pandemic contributed to the migration potential of doctors but was of secondary importance, our second hypothesis concerns the extent to which other, more long-term push factors, related to life and work in Poland, contribute to migration intentions:

H2: *Long-term factors related to the quality of work and life in Poland are more important for medical students and junior doctors considering emigration than the pandemic.*

**Mixed methods design.**    To gain a more comprehensive insight into the migration motives of medical students and junior doctors, the research is based on a mixed methods design and consists of two components complementing each other: an online survey among Polish medical students in their last year of studies (n = 205) and in-depth semi-structured interviews (n = 9), conducted several months later, when the respondents were already junior doctors, which were used to deepen the understanding of individuals' views and personal experiences, which cannot be explored with a questionnaire alone [70].

The survey was run in February 2022 at four large Polish medical universities: in Warsaw, Kraków, Gdańsk, and Białystok. These universities were selected for the study because they are public (i.e., offer studies free of charge) and gather the largest numbers of medical students in Poland. Also, thanks to their international certificates, graduation from these universities guarantees eligibility to work abroad. Prior to its release, the survey was consulted with the authorities of the universities and their student councils. To recruit participants, the survey link was sent by e-mail to 6th-year students by their universities. The survey was also advertised through the student councils' social media but these posts did not include the survey link to avoid obtaining responses from other people. The link led to a questionnaire on a University of Warsaw (UW) server, where the UW's Ankieter survey software was used, which guaranteed the security of the survey data.

The questionnaire consisted of 45 closed and open-ended questions. Respondents first had to express consent to participate. The first three questions were of a filtering nature. Respondents' demographic data was collected through 5 questions at the end. The survey was scheduled to take approximately 15–20 minutes.

The response rate was 15% (at the time of the questionnaire, 1366 students were enrolled in the 6th year of the Polish-language medical degrees in the four schools included). Among 205 questionnaires completed by Polish students (who were also born in Poland), 125 were completed by women and 80 by men, which reflects the gender balance among medical students and doctors in Poland. The responses were distributed as follows among the participating universities: Collegium Medicum Jagiellonian University in Cracow—CMUJ (34 out of 232 students enrolled); Warsaw Medical University—WUM (39 out of 643 students enrolled; Medical

University in Gdansk—GUMED (84 out of 276 students enrolled); Medical University in Bia-lystok—UMB (48 out of 215 students).

Nearly 57% of respondents were single and 97% did not have children. At the end of the questionnaire, respondents were asked if they could be contacted directly by the researchers in the future for an in-depth interview and subsequent stages of the survey. Those who chose to continue their participation (175 out of 205 respondents) provided e-mail addresses which could be used after graduation.

Nearly a year after the survey, between December 2022 and February 2023, selected respondents were interviewed and asked to reflect on the survey's key findings and the impact of the Covid-19 pandemic on doctors' migration plans. Respondents for the interviews were drawn from the list of persons who had provided e-mail addresses, with participants from each University listed in turn to ensure the participation of students from all schools, and the first participant drawn randomly from among the first nine. Interview requests were sent to every tenth survey respondent who had provided an e-mail address, and if the person did not respond to two e-mails, the following person on the list was recruited. As a result of this procedure, interview requests were sent out to 32 people. The requests resulted in 9 in-depth interviews with 6 men and 3 women, which were all conducted remotely (via Skype or ZOOM) and lasted, on average, 30 minutes. The gender balance of the respondents did not reflect the balance in the survey or among the student body. However, the responses in the survey did not differ significantly depending on gender. One participant had emigrated between the time of the survey and the interview, the others were working in the Polish public healthcare system. Consent for participation was obtained in writing before the interviews, and consent for recording at the beginning of the interview. The interviews were transcribed and anonymized. The research project has obtained the approval of the Research Ethics Committee of the Centre of Migration Research (nr CMR/EC/1/2022).

**Analysis & research findings.**    To assess the attitudes towards the probability of going abroad, an 11-point Likert scale was employed (Fig 1).

**Dependent variable.**    The survey question regarding migration intentions was: "*What is the probability of you going abroad to work*?". The 11-point Likert scale variable was split and recoded into dummy variable *mig1* (higher migration intentions) to account for those whose probability of going abroad was 6 and above on the Likert scale.

**Independent variable.**    The following independent variables were identified and included in the analysis:

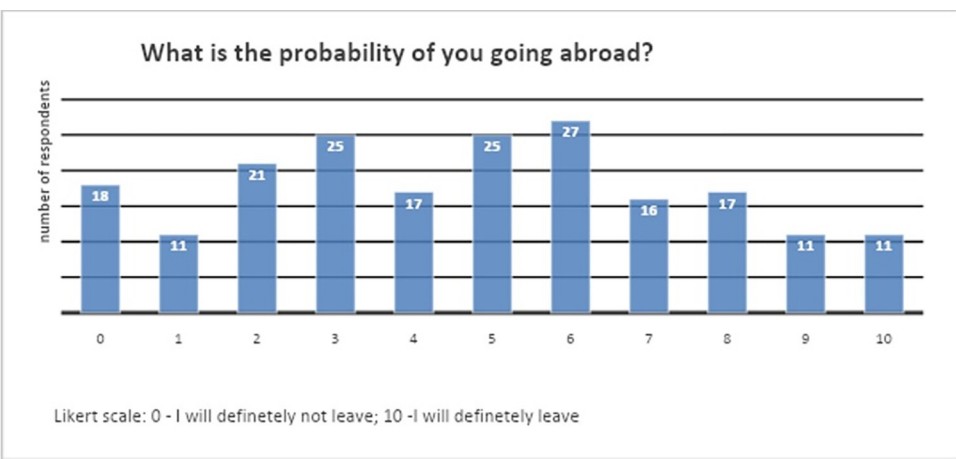

**Fig 1. Likert scale outcome of participants' probability of going abroad.**

1. The answers to the question: *"How did the experience of the Covid-19 pandemic affect the probability of emigration among Polish medical students?"*.

2. The second set of independent variables explored the question *"To what degree do the following factors push you to leave Poland?"*.

These are further referred to as *push factors*. They included the socio-political situation in Poland, housing situation, family situation, earnings, the organization of the public healthcare system in Poland, work conditions, career path, and working hours. The respondents could judge the influence of each of the above, including the influence of Covid-19, on a 5-point Likert scale from 1 - "not at all" to 5 - "to a large degree", with positive answers recoded into dummy variable 1. Kendall's tau-b correlation and logistic regression were used to test the associations between migration intentions and selected push factors.

The next analytical step presents the results of Kendall's tau-b correlation between the push factors and the perception of the influence of the Covid-19 experience as a contributing factor for the migration decisions of medical students. Based on selected push factors (determined by the results of Kendall's correlation) and the Covid-19 experience, the first logistic regression model was estimated. The 2nd model was further controlled for gender.

The second analytical component of this study is the results of the qualitative inquiry that were used to elaborate on the meaning of the results of the quantitative analysis. Participants' accounts addressed the experience of the Covid-19 pandemic, and how it affected their and their colleagues' migration plans. They also reflected and elaborated on other push factors identified in the estimated models.

**The Covid-19 experience.** Although most participants did not work in Covid wards, their indirect experiences shaped their perception and decision to remain in Poland or undertake migration. 63.5% agreed that the Covid-19 experience had significantly increased or increased the potential for medical students' migration decisions. 23% stated that the Covid-19 experience had no effect on the overall decisions to leave, and the remaining 13.5% believed that the pandemic experience decreased medical students' potential consideration of migration.

Table 1 presents the result of Kendall's rank correlation between the variable that in the participants' opinion the Covid experience increased the probability of emigration among medical students and the dependent variable of those who have higher migration intentions (mig1— those above the middle cut-off point: 0–5 and 6–10). The correlation indicates a positive moderate correlation between the two variables and the correlation coefficient is significant at the 5% level or lower.

**Push factors.** Participants could select various push factors influencing their migration intentions. Among them, especially the socio-political situation in Poland encouraged them to pursue migration plans. Additionally, factors related to work circumstances (working hours, work conditions, healthcare organization, followed by earnings) proved to be the most significant push factors (Fig 2).

Table 2 assesses the correlation between push factors and the group of higher migration intentions (Mig1). The push factors were grouped into two categories: related to medics' work circumstances and related to general circumstances. Regarding the general circumstances,

**Table 1. Kendall's tau-b correlation (n = 200).**

|  | Higher migration intentions |
| --- | --- |
| Covid experience increased migration intentions | 0.204* |

* Correlation coefficients significant at the 5% level or lower

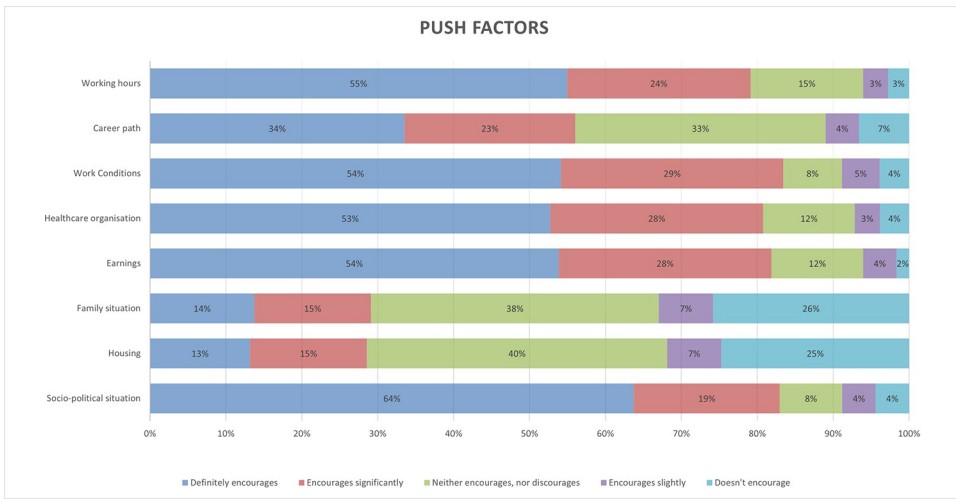

**Fig 2. Push factors encouraging medical students to leave Poland.**

both the socio-political situation and family situation have a positive moderate correlation and hold the strongest correlation among all the push factors included. The group of indicators related to work circumstances–career path and healthcare system organization–were relatively strongly correlated among other factors listed below.

Despite the overall belief that the Covid-19 experience has increased medical students' migration intentions, its effect weakens when confronted with other push factors. It, therefore, begs the fundamental question of whether the migration intentions of future doctors were driven by Covid-19, or was the pandemic experience only partly responsible for such decisions and the determinants of what encourages doctors to migrate are much more deeply embedded in the socio-political or work-related context in Poland.

**The Covid-19 pandemic and other push factors influencing emigration.** Logistic regression has further looked into the probability of selected push factors and general perception of the Covid-19 experience being likely to determine the migration plans. In line with previous results, the indicators related to the general circumstances of living in Poland remain much stronger than the Covid-19 experience. Similarly, some work-related factors also continue to have a higher probability of being an important push factor than the Covid-19 experience alone. Therefore, although Covid-19 alone seemed to strongly influence the intentions of

**Table 2. Kendall's rank correlation (n = 181).**

| Factors affecting migration decision | | Higher migration intentions |
|---|---|---|
| Covid-19 experience increased migration intention | | 0.076 |
| General circumstances | **Socio-political** | **0.208**[*] |
| | **Family situation** | **0.245**[*] |
| | Housing situation | 0.154[*] |
| Work circumstances | **Career path** | **0.175**[*] |
| | **Healthcare system organization** | **0.184**[*] |
| | Work conditions | 0.159[*] |
| | Working hours | 0.138[*] |
| | Earnings opportunity | 0.020 |

[*] Correlation coefficients significant at the 5% level or lower

**Table 3. Logistic regression of Covid-19 experience, selected push factors and migration intentions.**

|  | Model 1 |  | Model 2 |  |
|---|---|---|---|---|
|  | Coef | SE | Coef | SE |
| *Covid (mig.increase)* | .323 | .368 | .339 | .370 |
| *Career path* | .363** | .156 | .368** | .156 |
| *Healthcare system organization* | .375** | .188 | .376** | .189 |
| *Family situation* | .397** | .136 | .397** | .136 |
| *Socio-political* | .432** | .183 | .440** | .184 |
| *Gender (ref:men)* |  |  | -.307 | .356 |
| *pseudo R2* | 0.180 |  | 0.183 |  |
| *_cons* | -5.70*** | 1.43 | -5.58*** | 1.45 |

p<**0.10***,

p<0.05**

p<0.01***

migration among participants, when considered together with other push factors–its effect weakens. Further, controlling for gender, no statistically significant results have been found (Table 3).

Moreover, the most unequivocal answer in the whole survey concerned the statement "*Hate speech directed at doctors in Poland significantly increases their will to emigrate*", with which respondents were asked to agree or disagree on a 5-point Likert scale. 93% agreed or strongly agreed with this statement. They further elaborated on this issue in the qualitative interviews, as discussed below.

**Covid-19 vs. other factors—qualitative analysis.** The following section provides the outcome of the analysis based on qualitative interviews with selected survey participants (conducted several months after graduation, when most of the respondents were already junior doctors working in Polish hospitals), and the content of the open-ended questions in the survey. The results provide insight into how the pandemic and its management influenced the migration intentions of junior doctors and allow us to understand how this factor is related to other push factors.

In line with the results of the quantitative research above, the analysis of the qualitative interviews led us to the conclusion that the Covid-19 pandemic was not the main factor influencing migration intentions. However, for many respondents, it brought to light important push factors related to the organisation of the healthcare sector in Poland or the political atmosphere around doctors' work. Broadly speaking, the most important push factors mentioned as encouraging young doctors to leave Poland were of two categories—those related to work and those related to broader issues, not specific to doctors.

Young doctors were generally critical of the functioning of the healthcare system and particular hospitals. As one respondent summed up:

> *It is difficult to work in Polish hospitals because of overlapping competencies, poor use of equipment, and a lack of coordination between specialists. Often, it is about small things that significantly spoil the organization of work. This wastes a lot of time.* [Graduate 2, male]

Many respondents underlined that the level of investment in healthcare was too low. This influenced work conditions for doctors in terms of the buildings, equipment, number of medical tests, rehabilitation and other services they could prescribe to patients.

*The buildings and equipment are relics from the times when they were built. To take the patient for an ultrasound, you have to wheel him outside, into the cold, and to another pavilion.* [Graduate 3, male]

Interestingly, this overall level of investment and mismanagement of the healthcare sector seemed to be more significant than low wages, which have often been reported in Polish media as a problem and a cause of emigration. The opinions of medical students and junior doctors concerning earnings in Poland were divided, with some complaining about their inadequate level but others pointing out that compared to the costs of living, salaries are actually at similar levels as in Western Europe. On the other hand, they perceived investments into their training as inadequate, with problems ranging from an insufficient number of places in various specialisations and lack of funding for costly specialist courses to insufficient equipment or insufficient time senior colleagues could devote to their training due to staff shortages.

Many of the above-mentioned problems were, in the eyes of the respondents, exacerbated during the Covid-19 pandemic.

*I would like to say that we came out of it, but looking at hard data and the fact that Poland had the highest increase in deaths caused by the pandemic among EU countries, that the percentage of vaccinations is not satisfactory, it is somehow hard to say that we managed well.* [Graduate 3, male]

The pandemic was thus not seen as a decisive factor which influenced junior doctors' intentions to migrate but as a factor forcefully demonstrating existing problems.

*I wouldn't link it directly with Covid (. . .). Some people already plan to leave from the beginning of their studies. They then postpone these plans until those clinical years. It isn't linked with Covid, it was rather linked to other life plans.* [Graduate 1, female]

*From what I know from friends who worked in the Covid ward, they saw what life was like after university, what work was like (. . .) The decision to either leave or not leave was made beforehand and only reinforced. I doubt that the pandemic would have influenced such decisions.* [Graduate 8, male]

What is more, many respondents were discouraged by the political atmosphere around their work. Given that access to care was uneven, the anti-vaccination movement was quite strong, and the number of political scandals that surfaced during the pandemic was high, many respondents felt like their front-line jobs were not being taken seriously. At the same time, they believed that politicians and patients often blamed them for problems that were the result of the malfunctioning of the system. In interviews and open-ended questions of the survey, many pointed to the lack of societal understanding of doctors' work and mistrust, partly created by the rhetoric of politicians.

*People's attitudes, how society treats doctors. . . The lack of understanding of our work, and the fact that we are not the ones who are responsible for how this system works. Besides, working hours and the socio-political situation are the most encouraging reasons for people to leave.* [questionnaire, open-ended question]

*[Covid-19] didn't affect my migration plans, but it affected how I was perceived in the family, as a doctor. They saw me as part of a bigger big pharma conspiracy or so. . . some people really*

*disappointed me. Despite my trying to explain to them, my family really disappointed me.* [Graduate 2, male]

Hate speech from politicians, patients and on the internet was perceived as a particularly acute social problem.

*In the era of the pandemic, everyone has heard opinions about doctors, whether from politicians or other activists, that doctors don't want to work, that they are running away and so on. I think it may not even be that damaging to doctors, but it is just sad that someone working in their own country in the era of a pandemic hears that they don't want to work and that they are running away from duty. Well, I think if there was something to push someone abroad, if they were still hesitating over the decision whether to leave or not, then it might have helped them.* [Graduate 9, male]

As demonstrated above, junior doctors did not see the pandemic as a decisive push factor for migrations. However, they did see problems exacerbated by the pandemic—both organisational and in terms of political and public attitudes—as serious and possibly acting as an additional factor reinforcing migration decisions of those who had already been considering migration once they reached the appropriate moment in their career (e.g. after completing their internship, after obtaining a specialisation).

Additional political factors were also seen as making doctors' work more difficult. This was particularly related to abortion. Abortion regulations were further tightened in Poland through a court decision in 2020, which led to the media reporting cases of pregnant women dying when they could have perhaps been saved by an abortion. Doctors felt that they would be blamed, irrespective of what decisions they made.

*For me, there is such a lack of certainty when it comes to what the future will look like and how the doctor will be treated in Poland and what the legal threat will be towards the doctor. The first example, from casual conversations, is that fewer and fewer people are interested in gynecology and obstetrics. It used to be a popular specialisation, but nowadays people don't want to go into it, they are afraid.* [Graduate 9, male]

Last but not least, socio-political factors not related to the work of doctors were also mentioned as significant for some participants. This concerned, for example, the treatment of LGBTQ people—both the lack of marriage and adoption rights and the homophobic rhetoric of politicians in government at the time.

The above analysis confirms that factors related to the organization of healthcare and socio-political factors were the most significant in pushing junior doctors to leave. The Covid-19 pandemic in itself did not change the plans of our respondents. However, in their eyes, it highlighted and exacerbated existing problems related to the organisation of healthcare and the treatment of doctors by the authorities and by other members of society. For those who were already considering migration, it may have thus acted as a trigger or last straw.

## Discussion

The quantitative and qualitative analysis above has demonstrated the importance of factors related to healthcare, such as its organisation and working hours, the possible career paths of doctors, or equipment, as influencing the migration intentions of medical students and junior doctors. What is more, socio-political factors, such as hate speech directed at doctors, proved even more important. Our results expand previous scholarships on the importance of

emigration among junior doctors as an alternative career path. In previous empirical studies, researchers have mainly identified factors related to the quality of employment and living conditions [1, 25], but the influence of socio-political factors on junior doctors' emigration intentions has not been noticed. Of particular importance is the factor of hate speech towards a particular professional group. Our research has shown that junior doctors may be encouraged to emigrate by unfriendly attitudes from patients and politicians, as well as disrespect for basic rights, including the right to live according to one's beliefs. The increase in resentment towards doctors observed during the pandemic in Poland may reinforce their decisions to move abroad. This finding is highly relevant to the discussion of 'push/retain' and 'pull/repel' factors [33, 38]. It introduces a new element into the discussion on the determinants of high-skilled migration, extending the list of possible push factors that need to be considered. The research tests in practice to what extent unexpected circumstances have a real impact on migration decisions, or whether they are only of secondary importance.

During the implementation of the survey, limitations were identified that may affect the results to a certain extent, especially of the quantitative survey. Most respondents had no experience of working in Covid wards, and so they drew their knowledge from colleagues who did have such experience. At the same time, most of them had had contact with Covid-19 patients as part of their internship or learning experience in hospitals. They were therefore able to observe how the pandemic was being dealt with by the Polish healthcare system. A certain limitation was also that the study was conducted while the pandemic was still ongoing, which did not allow the respondents to gain some distance before judging how the Polish healthcare system coped with this emergency. Since the respondents of the survey were medical school students, their perceptions of various factors may change significantly in the first years of their careers as doctors, so the study should not be treated as a prediction of migration levels but rather as exploring factors which influence these decisions.

The survey was discontinued on 26 February 2022, somewhat ahead of schedule, to avoid the risk that the outbreak of war in Ukraine would affect declared migration plans. This resulted in a lower response rate but we felt that it was necessary, since the atmosphere of looming direct threat, present in Poland in the first days after the Russian attack on Ukraine could have influenced responses.

## Conclusions

The article aimed to establish to what degree the Covid-19 pandemic and its management influenced the migration intentions of Polish medical students and junior doctors. We found that the difficulties of the Polish healthcare system in dealing with the pandemic had a slightly positive influence on these intentions. At the same time, this factor was secondary–its occurrence was not decisive in the intentions to emigrate but reinforced earlier plans to go abroad. Factors such as the socio-political situation in Poland, participants' family situation, issues related to hate speech or the perception of doctors' competence, employment conditions or the possibility of pursuing a planned career path proved to be much more important. Interestingly in our study, salary did not turn out to be the most important factor. This allows us to conclude that in the case of medical students migration decisions have a very well-established basis that does not fundamentally change even under the influence of such a dramatic situation as the pandemic. This conclusion has important implications for the ongoing discussion on the evolution of push and pull factors in place and time as well as the importance of the impact of extraordinary events or circumstances (such as a pandemic) on the migration decisions of high-skilled personnel.

To conclude, given the importance of factors relating to living and working conditions in Poland as revealed by the results of the qualitative and quantitative study, H1 (*The Covid-19*

*pandemic has increased the migration intentions of junior doctors in Poland)* is only partially confirmed. Although the experience of the Covid-19 pandemic exposed the weakness of the Polish healthcare system, the migration intentions of medical students and junior doctors in Poland were also influenced by other factors related to the safety and quality of work and life in Poland, some of which became more pertinent due to the pandemic experience. Those other factors turned out to be not only equally important as the Covid-19 experience, but far more significant, as suggested in H2 (*Long-term factors related to the quality of work and life in Poland are more important for junior doctors considering emigration than the pandemic).*

The Covid-19 pandemic partly co-occurred in Poland with another phenomenon which affected the daily lives of people, including medical students and junior doctors: Russia's aggression against Ukraine, which resulted in a massive arrival of war refugees and introduced another element of uncertainty. It is an open question to what extent the war in Ukraine, with the Covid-19 pandemic still ongoing, albeit with less intensity, will increase the emigration potential of junior doctors. A positive or negative answer to the question of the impact of the war in Ukraine on migration intentions would be a further contribution to the discussion on the evolution of push and pull factors in space and time under the influence of sudden events of mass character and scope. This issue will be addressed by the research team that authored this paper.

## Author Contributions

**Conceptualization:** Dominika Pszczółkowska, Maciej Duszczyk, Kamil Matuszczyk.

**Data curation:** Sara Bojarczuk.

**Formal analysis:** Sara Bojarczuk.

**Funding acquisition:** Maciej Duszczyk.

**Investigation:** Maciej Duszczyk, Kamil Matuszczyk, Emilia Szyszkowska.

**Methodology:** Dominika Pszczółkowska, Maciej Duszczyk.

**Project administration:** Dominika Pszczółkowska.

**Supervision:** Maciej Duszczyk.

**Writing – original draft:** Sara Bojarczuk, Kamil Matuszczyk, Emilia Szyszkowska.

**Writing – review & editing:** Dominika Pszczółkowska.

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
