## [Decision Letter · Decision Letter 0]

24 Jul 2023

PONE-D-23-08865Did Covid-19 make things worse? The pandemic as a push factor stimulating the emigration of young doctors from Poland: a mixed methods studyPLOS ONE

Dear Dr. Pszczółkowska

Thank you for submitting your manuscript to PLOS ONE. After careful consideration, we feel that it has merit but does not fully meet PLOS ONE’s publication criteria as it currently stands. Therefore, we invite you to submit a revised version of the manuscript that addresses the points raised during the review process.

The article needs very serious changes. I fully agree with the reviewers' comments. Unfortunately, all elements of the text need to be improved, so the text would basically need to be rewritten. Of the elements I would particularly like to emphasize is the need to improve and update the literature review as well as build a narrative around the research problem and hypotheses. In doing so, please pay particular attention to the terms you use, and their conceptualization and consistency in using them. The entire methodology section needs to be improved so that it is clearer, more transparent and allows the reader to understand what has been done and thus enable replication of the research. 

Please separate discussion from results and from conclusion, include limitations and most importantly theoretical and managerial contribution.

The manuscript needs a lot of work And, unfortunately, it is difficult at this point to promise that it will be publishable once the text is improved. However, we see potential in it, so we hope that the authors will nevertheless attempt to improve the text, as the topic is very important and the research carried out is relevant.

We look forward to receiving your revised manuscript.

Kind regards,

Jolanta Maj

Academic Editor

PLOS ONE

Journal Requirements:

**When submitting your revision, we need you to address these additional requirements.**

**1. Please ensure that your manuscript meets PLOS ONE's style requirements, including those for file naming. The PLOS ONE style t**emplates can be found at

Additional Editor Comments (if provided):

The article needs very serious changes. I fully agree with the reviewers' comments. Unfortunately, all elements of the text need to be improved, so the text would basically need to be rewritten. Of the elements I would particularly like to emphasize is the need to improve and update the literature review as well as build a narrative around the research problem and hypotheses. In doing so, please pay particular attention to the terms you use, and their conceptualization and consistency in using them. The entire methodology section needs to be improved so that it is clearer, more transparent and allows the reader to understand what has been done and thus enable replication of the research.

Please separate discussion from results and from conclusion, include limitations and most importantly theoretical and managerial contribution.

The manuscript needs a lot of work And, unfortunately, it is difficult at this point to promise that it will be publishable once the text is improved. However, we see potential in it, so we hope that the authors will nevertheless attempt to improve the text, as the topic is very important and the research carried out is relevant.

Reviewers' comments:

Reviewer's Responses to Questions

**Comments to the Author**

1. Is the manuscript technically sound, and do the data support the conclusions?

Reviewer #1: No

Reviewer #2: Partly

2. Has the statistical analysis been performed appropriately and rigorously? 

Reviewer #1: No

Reviewer #2: I Don't Know

3. Have the authors made all data underlying the findings in their manuscript fully available?

Reviewer #1: Yes

Reviewer #2: No

4. Is the manuscript presented in an intelligible fashion and written in standard English?

Reviewer #1: No

Reviewer #2: No

5. Review Comments to the Author

Reviewer #1: Thank you for providing me the opportunity to offer my suggestions to the authors for this article on the topic “Did Covid-19 make things worse? The pandemic as a push factor stimulating 14 the emigration of young doctors from Poland: a mixed methods study”. The study is interesting but may need some revision.

1. I am suggesting that the abstract should be structured; introduction, aims, methods, findings & conclusions. Please, kindly present significant findings and for the quantitative aspect give us the exact figures in the abstract.

2. In the introduction section the authors made a case for this study. However certain claims were made which may require some revisions

i. In line 65-68, the authors made assumption….. “Thus, we assume the experience of the Covid-19 pandemic, in particular, the response of the given health system and the working conditions for doctors may influence the decision to leave the country of residence”

ii. They went further make inferences based on the assumptions. I am suggesting these information could have be captured well enough.

3. A do not know if the literature review is necessary. Perhaps all those information could be summarized to make up the introduction.

4. I see some repetition in the work, I am suggesting to the authors to avoid repetition in the entire work.

5. The entire method secure is scanty and does not give vivid decriptio of how the study was done.

I. I am suggesting some links for you to read and consider a revision of the entire methods sections.

II. https://www.bmj.com/content/371/bmj.m4435#:~:text=Mixed%20methods%20research%20designs%20have,%2C%20implementation%2C%20and%20reporting%20stages.

III. https://journals.sagepub.com/doi/full/10.1177/1558689819875832

IV. You can also visit the STROBE for checklist on observational studies (https://www.strobe-statement.org/checklists/ )

6. I can see the authors compared the results to discussion straight away in one headings.

7. This is making the results difficult to read through

8. The discussion does not actually provide adequate evidence from previous work. No adequate reasoning is provided to enable readers contextualize the key findings.

9. I am suggesting to the authors to consider writing the results separated from the discussion

10. The conclusion is too much. Please summarized

11. Include the limitation and strength

Overall, the manuscript is good but would need extensive revision. Thank you

Reviewer #2: Thank you for this interesting submission. Unfortunately, I feel as though it needs further work in order to be ready for publication. At the moment, certain terms and elements of the analysis remain unclear. For instance, the authors refer to 'young doctors' then 'new doctors' and then 'student doctors' 'medical graduates' 'medical students' 'future doctors' - please make this uniform and consistent so as not to confuse the reader. Perhaps a definition would help. Same goes for 'migration decisions' versus 'migration potential' or 'migration intentions' - these are not the same and so should not be used interchangeably (e.g. line 71 & 75). The analysis section needs further clarification i.e. it is unclear whether H2 was or was not confirmed; we don't know what 'qualitative analysis' was employed; details of respondents (at least the qual part) need to be provided to contextualise analysis, especially as issues related to gender are explored. I think that currently the paper reads as quite shallow in its analysis and the literature review does not do it any justice as it's not organized in a coherent manner to support the findings (e.g. line 90-91 - quite an obvious simplistic statement). The paper would benefit from a more structured debate and a closer proofread as in places it is difficult to follow the line of thought e.g. line 60-61; 278; 336. More consistency around tenses would be good. Some references are either quite old or not there e.g. the very first sentence of the introduction refers to a study from 2006; line 67, 232, 234, 264, 280 - missing references. Additionally, although the authors note 'this conclusion/findings have important implications for the ongoing discussion' they don't actually explain this statement or support it with some adequate examples as to how this is the case. I'm sorry if this seems as harsh but I do hope that this review will be helpful in strengthening this paper.

6. PLOS authors have the option to publish the peer review history of their article (what does this mean?). If published, this will include your full peer review and any attached files.

Reviewer #1: No

Reviewer #2: No

---

## [Author Response · Author response to Decision Letter 0]

15 Sep 2023

Dear Editor, 

Dear Reviewers,

Thank you very much for your work and comments, which we believe have helped us significantly improve the paper, particularly its clarity and structure. Here is a summary of the changes made, in the order in which they appear in the paper:

1. The abstract has been edited and quantitative results have been added (in line with the suggestion of Reviewer 1). 

2. Our main terms are used consistently throughout the abstract and paper and are defined in the conceptual section. In line with migration literature, we speak of “migration intentions”, as well as “medical students” and “junior doctors”. We have decided that “junior” is a better translation than the previously used “young”, as a term that is more frequent and precisely defined in the academic literature in line with what we mean – persons who have already graduated from medical school but have not completed their specialisation and reached consultant or general practitioner level (in line with the suggestions of the Editor and Reviewer 2).

3. The whole first part (up to page 12) has been rewritten with the following aims in mind (in line with the suggestions of the Editor and both Reviewers): 

a) to update the literature review and focus the narrative even more on the topic of factors pushing doctors to migrate, and on what is already known about the influence of Covid-19. Some other information regarding the situation of the health sector in Poland has been deleted, as not directly relevant;

b) to avoid repetitions. 

4. Much detailed information has been added regarding our methods and respondents. We have also placed the data used in this study in a repository (OSF - https://osf.io/vkbnx/), so that the study may be replicated by other authors. (Editor and Reviewer 1) The section based on qualitative data has been rearranged to focus it more on our main topic.

5. The discussion and conclusions sections have been separated, and the conclusions synthesized (Reviewer 1). Limitations are dealt with at the end of the discussion (Reviewer 1). We have also made clearer what the theoretical contribution of our work is (Editor and Reviewer 2). 

6. Finally, we have also used this opportunity to proofread the paper and make sure all references are in place. 

Again, thank you for your time and helpful comments. We hope that the manuscript is now much more transparent, making our contribution clear.

Best regards,

Dominika Pszczółkowska 

(on behalf of all the authors)

---

## [Decision Letter · Decision Letter 1]

2 Nov 2023

PONE-D-23-08865R1Did Covid-19 make things worse? The pandemic as a push factor stimulating the emigration intentions of junior doctors from Poland: a mixed methods study

PLOS ONE

Dear Dr. Pszczółkowska, 

Thank you for submitting your manuscript to PLOS ONE. After careful consideration, we feel that it has merit but does not fully meet PLOS ONE’s publication criteria as it currently stands. Therefore, we invite you to submit a revised version of the manuscript that addresses the points raised during the review process.

I concur with the assessments provided by the two reviewers. Their insights and feedback have been invaluable in evaluating the submitted manuscript. It is clear that the manuscript has strong potential, but there are certain methodological aspects that warrant further clarification and refinement. Addressing these concerns will undoubtedly enhance the quality and rigor of the research. I encourage the authors to carefully consider the reviewers' suggestions and make the necessary revisions to ensure that the methodology is well-defined and robust.

We look forward to receiving your revised manuscript.

Kind regards,

Jolanta Maj

Academic Editor

PLOS ONE

Reviewers' comments:

Reviewer's Responses to Questions

**Comments to the Author**

1. If the authors have adequately addressed your comments raised in a previous round of review and you feel that this manuscript is now acceptable for publication, you may indicate that here to bypass the “Comments to the Author” section, enter your conflict of interest statement in the “Confidential to Editor” section, and submit your "Accept" recommendation.

Reviewer #2: (No Response)

Reviewer #3: (No Response)

2. Is the manuscript technically sound, and do the data support the conclusions?

Reviewer #2: Partly

Reviewer #3: Yes

3. Has the statistical analysis been performed appropriately and rigorously? 

Reviewer #2: I Don't Know

Reviewer #3: Yes

4. Have the authors made all data underlying the findings in their manuscript fully available?

Reviewer #2: Yes

Reviewer #3: Yes

5. Is the manuscript presented in an intelligible fashion and written in standard English?

Reviewer #2: No

Reviewer #3: Yes

6. Review Comments to the Author

Reviewer #2: Thank you for this revised submission, it has much improved, yet, I feel it needs further work to make it of a publishable standard.

Here are my main concerns:

1. It is uncommon to have empirical contribution stated in the introduction, better to move it to conclusion.

2. Language needs tightening, it feels rushed e.g.:

100 'good ...' - what?

102 'destination' country?

284 'doctors' - meaning the participants?

451 which previous scholarship?

462 how?

decide whether to use LGBT or LGBTQ or LGBTQ+

3. The paper seems difficult to follow at times, e.g.:

330-331; 343 - yes, and what did they say?

4. The qualitative part i.e. the interview extracts seem to be just described in the analysis section - state what is particularly interesting and relevant to healthcare staff as it appears that most of what is reported could apply to the general public too.

5. The discussion section doesn't offer a discussion - a synthesis of the results is needed here.

6. The authors white of 'socio-political factors in the context of junior doctors' emigration intentions' - but most of the participants don't intend to emigrate, right? If so, then how is this the 'context'? Perhaps this should be rephrased as at the moment it seems that the paper is about covid's 'impact on migration decisions' but did they even consider emigration before they were asked about it?

Reviewer #3: A key strength of this study is the use of mixed methods with the same participant group, providing both quantitative data on migration intentions and qualitative insights into reasons behind them. However, the survey methodology requires clarification. More details are needed on survey response rate and representativeness. How many participants provided contact info for interviews and what was the response rate? Without this information, it is difficult to assess potential nonresponse or selection bias.

The authors should also clarify inconsistent labeling of the interview participants as both "medical students" and "junior doctors." It is applicable to the results section: the authors, when analyzing the quantitative survey results (lines 317-319), used the concept of "junior doctors" while the participants were still medical students at the time of the research.

Regarding generalizability, the lack of information on survey respondent distribution across different universities is a limitation. As social media recruitment was used, respondents beyond the targeted university may have participated without the authors' knowledge, hampering generalizability. Further details on number of respondents by university year would also be beneficial. The small interview sample raises similar concerns, as reasons for selecting 6 men and 3 women are unclear given gender distribution differences in the medical student/doctor population.

While the study provides useful exploratory data, the limitations hinder broad applicability of the conclusions. I suggest revising the framing to focus more precisely on this cohort rather than generalizing. Additional details on sampling and methodology are needed to assess rigor and potential biases. Discussion of study limitations should also be expanded. Overall, this is an interesting research question but requires refinement to match the appropriate scope of inference.

More detail suggestions:

Defining key terms like “migration intention,” “medical students,” and “junior doctors” should be moved to the introduction or methods section.

Line 169 – bracket needed

7. PLOS authors have the option to publish the peer review history of their article (what does this mean?). If published, this will include your full peer review and any attached files.

Reviewer #2: No

Reviewer #3: No

---

## [Author Response · Author response to Decision Letter 1]

18 Dec 2023

Dear Editor, Dear Reviewers,

Please refer to the file "Response to Reviewers 2" where we respond to all comments.

Best regards,

The Authors

---

## [Decision Letter · Decision Letter 2]

27 Feb 2024

PONE-D-23-08865R2Did Covid-19 make things worse? The pandemic as a push factor stimulating the emigration intentions of junior doctors from Poland: a mixed methods studyPLOS ONE

Dear Dr. Pszczółkowska,

Thank you for submitting your manuscript to PLOS ONE. After careful consideration, we feel that it has merit but does not fully meet PLOS ONE’s publication criteria as it currently stands. Therefore, we invite you to submit a revised version of the manuscript that addresses the points raised during the review process.

Myself and the reviewers are very grateful for implementing the suggested changes in the text. In our opinion, the modifications have indeed addressed most of the crucial remarks. Nevertheless, there are still a few minor issues in the text that need further consideration, incorporation, or addressing, as indicated by the reviewers.

We look forward to receiving your revised manuscript.

Kind regards,

Jolanta Maj

Academic Editor

PLOS ONE

Journal Requirements:

Reviewers' comments:

Reviewer's Responses to Questions

**Comments to the Author**

1. If the authors have adequately addressed your comments raised in a previous round of review and you feel that this manuscript is now acceptable for publication, you may indicate that here to bypass the “Comments to the Author” section, enter your conflict of interest statement in the “Confidential to Editor” section, and submit your "Accept" recommendation.

Reviewer #2: (No Response)

Reviewer #3: All comments have been addressed

2. Is the manuscript technically sound, and do the data support the conclusions?

Reviewer #2: Yes

Reviewer #3: Yes

3. Has the statistical analysis been performed appropriately and rigorously? 

Reviewer #2: I Don't Know

Reviewer #3: Yes

4. Have the authors made all data underlying the findings in their manuscript fully available?

Reviewer #2: No

Reviewer #3: Yes

5. Is the manuscript presented in an intelligible fashion and written in standard English?

Reviewer #2: Yes

Reviewer #3: Yes

6. Review Comments to the Author

Reviewer #2: Thank you for this revised submission. It has much improved, although, I still have a few issues with it:

1. 'Migration decisions have a very well established basis' - this is a very problematic statement in my view, which is due to existing scholarship which says otherwise and also due to how definitive it is. That said, I'm not here to argue my point of view and it may just be that, a point of view.

2. line 83 - 'pandemics' reads awkward; also, did I miss this lit review of the influence of health crisis the authors refer to? Additionally, this contradicts line 109 where it says that the review only includes publications from the ages 2009-2019 which is pre-covid.

3. line 178 - 'influx' is a negative word with pejorative connotations and thus I wouldn't use it when describing those seeking asylum due to an armed conflict - there's a debate on the new acceptable terms within migration studies that the authors may wish to follow.

4. I'd be very interested to learn the ages of the respondents but I recognize that this isn't necessary for this submission but it would give more context and enable better understanding of the results.

5. line 372 - 'resumed' ?

6. line 441 - it led to spreading the word/publicizing details in national media about women dying due to a lack of access to legal abortion and not to publicized cases

7. line 470 - I wouldn't call it 'under-recognized factor of hate speech' - see a list of available resources that came up through just a quick search:

https://blogs.bcm.edu/2023/07/14/hate-speech-in-healthcare/

https://czasopisma.uwm.edu.pl/index.php/mkks/article/view/6410

https://www.mirecc.va.gov/visn16/working-with-patients-who-use-hate-speech.asp

https://www.taylorfrancis.com/chapters/edit/10.4324/9780429201813-6/bad-bedside-manner-sheri-wells-jensen-claire-wells-jensen

Reviewer #3: As I have reviewed the article in the first round, and had paid attention to necessary changes, which were incorporated by the authors in the subsequent submission, there are not too many comments on the reviewed text.

However, there are some suggestions:

It would be advisable to emphasize more clearly the justification for the necessity of conducting research in Poland – it is worthwhile to refer to the OECD report, providing specific indicators of the number of doctors/nursing staff per 1000 inhabitants in Poland.

There is a question about conducting a literature review for the years 2009-2019? (10-years period?).

There is a lack of a clear message/key presentation in the introduction of the literature review. Perhaps it is worth considering a division based on push and pull factors, factors related to different disciplines.

7. PLOS authors have the option to publish the peer review history of their article (what does this mean?). If published, this will include your full peer review and any attached files.

Reviewer #2: No

Reviewer #3: No

---

## [Author Response · Author response to Decision Letter 2]

8 Mar 2024

Dear Reviewers,

Please find our response and a detailed description of the changes made in the attached file "Response to Reviewers with version 4".

Best regards,

The authors

---

## [Decision Letter · Decision Letter 3]

22 Mar 2024

Did Covid-19 make things worse? The pandemic as a push factor stimulating the emigration intentions of junior doctors from Poland: a mixed methods study

PONE-D-23-08865R3

Dear Dr. Pszczółkowska

We’re pleased to inform you that your manuscript has been judged scientifically suitable for publication and will be formally accepted for publication once it meets all outstanding technical requirements.

Kind regards,

Jolanta Maj

Academic Editor

PLOS ONE

Additional Editor Comments (optional):

After thorough consideration of the manuscript and the feedback provided by the reviewers, we have decided to accept the paper for publication, despite one of the previous reviewers declining to review the paper again.

Although one of the reviewers declined to participate in the review process again, it is noteworthy that the other reviewer, who has reviewed the paper for the third time, has accepted the paper. In their latest review, the reviewer expressed satisfaction with the revisions made by the authors and recommended acceptance of the manuscript.

Furthermore, upon careful examination, I found that the changes requested by the reviewer who declined to review again, following the previous round of minor revisions, have been adequately addressed by the authors. As an editor with expertise in the subject matter, I am confident that the manuscript now meets the standards of the journal in terms of clarity, methodology, and scholarly contribution.

Given the positive assessment from one reviewer and my own evaluation of the manuscript, I believe that inviting a new reviewer at this point may not significantly add to the review process. The revisions have been diligently implemented, and the paper now stands ready for publication.

I would like to extend my appreciation to the reviewers who have contributed their expertise to the review process and to the authors for their commitment to improving the manuscript.

Reviewers' comments:

Reviewer's Responses to Questions

**Comments to the Author**

1. If the authors have adequately addressed your comments raised in a previous round of review and you feel that this manuscript is now acceptable for publication, you may indicate that here to bypass the “Comments to the Author” section, enter your conflict of interest statement in the “Confidential to Editor” section, and submit your "Accept" recommendation.

Reviewer #2: All comments have been addressed

2. Is the manuscript technically sound, and do the data support the conclusions?

Reviewer #2: Partly

3. Has the statistical analysis been performed appropriately and rigorously? 

Reviewer #2: I Don't Know

4. Have the authors made all data underlying the findings in their manuscript fully available?

Reviewer #2: No

5. Is the manuscript presented in an intelligible fashion and written in standard English?

Reviewer #2: Yes

6. Review Comments to the Author

Reviewer #2: (No Response)

7. PLOS authors have the option to publish the peer review history of their article (what does this mean?). If published, this will include your full peer review and any attached files.

Reviewer #2: No

---

## [Editor Report · Acceptance letter]

2 Apr 2024

PONE-D-23-08865R3 

PLOS ONE

Dear Dr. Pszczółkowska, 

I'm pleased to inform you that your manuscript has been deemed suitable for publication in PLOS ONE. Congratulations! Your manuscript is now being handed over to our production team.

Kind regards, 

on behalf of

Dr. Jolanta Maj 

Academic Editor

PLOS ONE